# Skyrmion pinning energetics in thin film systems

Raphael Gruber [1], Jakub Zázvorka [2], Maarten A. Brems [1], Davi R. Rodrigues [1,3,4], Takaaki Dohi [1], Nico Kerber [1], Boris Seng[1,5], Mehran Vafaee[1,6], Karin Everschor-Sitte [1,4,7], Peter Virnau[1] & Mathias Kläui [1✉]

A key issue for skyrmion dynamics and devices are pinning effects present in real systems. While posing a challenge for the realization of conventional skyrmionics devices, exploiting pinning effects can enable non-conventional computing approaches if the details of the pinning in real samples are quantified and understood. We demonstrate that using thermal skyrmion dynamics, we can characterize the pinning of a sample and we ascertain the spatially resolved energy landscape. To understand the mechanism of the pinning, we probe the strong skyrmion size and shape dependence of the pinning. Magnetic microscopy imaging demonstrates that in contrast to findings in previous investigations, for large skyrmions the pinning originates at the skyrmion boundary and not at its core. The boundary pinning is strongly influenced by the very complex pinning energy landscape that goes beyond the conventional effective rigid quasi-particle description. This gives rise to complex skyrmion shape distortions and allows for dynamic switching of pinning sites and flexible tuning of the pinning.

[1] Institute of Physics, Johannes Gutenberg-Universität Mainz, Staudingerweg 7, Mainz 55128, Germany. [2] Institute of Physics, Faculty of Mathematics and Physics, Charles University, Ke Karlovu 5, Prague 12116, Czech Republic. [3] Dipartimento di Ingegneria Elettrica e dell'Informazione, Politecnico di Bari, Via E. Orabona 4, Bari 70125, Italy. [4] Faculty of Physics, University of Duisburg-Essen, Lotharstraße 1, Duisburg 47057, Germany. [5] Institut Jean Lamour, UMR CNRS 7198, Université de Lorraine, 2 allée André Guinier, Nancy 54011, France. [6] Singulus Technologies AG, Hanauer Landstraße 103, Kahl am Main 63796, Germany. [7] Center for Nanointegration Duisburg- Essen (CENIDE), University of Duisburg-Essen, Carl-Benz-Straße 199, Duisburg 47057, Germany. ✉email: klaeui@uni-mainz.de

Magnetic skyrmions are topologically stabilized chiral magnetic structures which are stabilized by bulk or interfacial Dzyaloshinskii-Moriya interaction (DMI)[1–3]. Due to their quasi-particle nature, skyrmions are of special interest in research for their promising applications[4–8]. In particular, skyrmions are suggested as information carriers in data storage devices[9,10] and logic devices[6,8,11,12] since they can be moved deterministically and efficiently by ultralow current densities due to spin-orbit and spin-transfer torques. Recently, the diffusive motion of skyrmions due to thermal excitations was demonstrated in thin-film multilayers which makes them applicable for nonconventional computing[13–16] and leads to 2D phases and phase transitions[17,18].

While for strongly driven fast skyrmion motion, pinning effects are expected to be small[19], pinning effects have been shown to play a crucial role in most skyrmion-based spintronic device proposals. For racetrack memories, for example, pinning may be used as notches to maintain a fixed distance between skyrmions[12,20]. Also nonconventional computing proposals such as reservoir computing require strong pinning of the skyrmions[21,22]. The key step to realizing such nonconventional computing in skyrmion textures is to obtain the appropriate pinning strength for which one needs a method to quantify and then understand the pinning effects. Moreover, in applications, which rely on skyrmion diffusion, such as Brownian computing and probabilistic computing, pinning effects are of crucial importance as the pinning strength is often comparable to the scales of thermal excitation and thus impacts the operation of skyrmion-based devices[14,15]. Pinning or more precisely a nonflat energy landscape for skyrmions occurs in skyrmion systems where different interactions on similar energy scales appear. Pinning effects due to local changes in the material parameters compete with magnetic interactions and have a strong impact on the static and dynamic skyrmion properties[4] such as the shape and profile of skyrmions in the static ground state and during motion[23,24]. In particular also properties of skyrmion motion and dynamic effects such as the skyrmion Hall effect[24,25] are strongly affected by the distribution of pinning sites leading potentially to different skyrmion Hall angle dependences on the skyrmion size[26,27]. The local material inhomogeneities leading to local variations of the magnetic parameters define an energy landscape[23] which can qualitatively impact the dynamics depending on the details of the pinning[14,24,26].

Skyrmion pinning in general is an open question and hence, quantifying and understanding the nature of the pinning experimentally is crucial[23]. So far there have been only first attempts to characterize pinning strengths by measuring the driving forces required to unpin the skyrmion from a specific point in space[28]. Furthermore, energy landscape dependences on pinning effects have been studied, but limited to skyrmions of sizes similar to the grains of the sample[22,29,30].

In particular, the current understanding of pinning is primarily based on micromagnetic simulations or on theoretical predictions considering effective potentials or the rigid particle description of skyrmions[23,31–39]. Several mechanisms have been considered to be responsible for pinning effects and give plausible origins for the wide range of pinning strength occurring in various materials at different external parameters[23]. These mechanisms include local changes of the magnetic parameters induced by variations in anisotropy[31], exchange interaction[32,33] or DMI[34] and the impact of missing spins, atomic impurities[35] or surface adatoms[36,37]. Depending on the mechanism, defects attracting[32,35–38] and repelling[35–38] the skyrmion center or even exhibiting a combined behavior of attraction and repulsion[36,37] have been predicted and established the conventional picture of skyrmion center pinning. Thereby, the skyrmion center has been used to describe the

position of the skyrmion with respect to a pinning site as centered or off-center. These theoretical predictions have been based on micromagnetic simulations[31–34,38], first principle calculations[35–37] or the equations of motion for a quasi-particle model[39] with partly contradicting claims about the mechanism of the pinning. Even though previous investigations have noticed an influence of deformations of skyrmions, a thorough investigation and solid understanding of the mechanism is still missing[30,33,38]. Experimentally, little work is available with in particular strong pinning reported in thin films[28,29,40–43] where explanations for pinning have been based on grain boundaries that are expected to occur in the polycrystalline films and thus pinning has so far always been considered to be a static property of the sample. However, these predictions and possible explanations have not been explored in detail experimentally. In particular, the effective potential or rigid quasi-particle descriptions typically used for skyrmions have neglected their complex spin structure with a core as well as a delineating domain wall boundary both being deformable and do not allow for a description of many of the complex pinning properties observed experimentally.

In this paper, we experimentally explore the skyrmion pinning behavior present in a thin film system. We develop a method to ascertain the energy landscape for skyrmions quantitatively and, by magnetic microscopy we reveal the details of the spatially resolved skyrmion pinning. We show in our analysis of the pinning mechanism that one has to go beyond a simplified quasi-particle model as skyrmion pinning arises from the domain walls allowing us to understand the complex skyrmion pinning observed in multilayer samples and we demonstrate that skyrmion pinning can be tuned on-the-fly by switching certain pinning sites on and off.

## Results

**Skyrmion energy landscape**. We start by probing the spatial dependence of the skyrmion pinning strength in a Ta(5)/$Co_{20}Fe_{60}B_{20}$(1)/Ta(0.08)/MgO(2) sample (thicknesses in nm, details for sample fabrication in the methods section) as the first important piece of information necessary to understand the pinning is to ascertain the energy landscape. To achieve this, we make use of the unique thermal dynamics of skyrmions that leads to a diffusive random walk motion that explores the full space of the sample[14]. We measure spatially resolved the occurrence (dwell time in relation to the full measurement time) and plot in Fig. 1a the probability of pixels being occupied by skyrmions throughout the measurement time (for details, see methods section). These results thus reflect the probability of finding a skyrmion covering a certain area of the sample.

We notice that at specific positions, skyrmions are found over nearly the entire observation time despite the regular re-nucleation procedures in between the single videos (see methods for details). These positions are characterized by localized high peaks of the probability density corresponding to skyrmions being strongly pinned. Other pinning sites show a weaker effect on skyrmions, meaning that they pin the skyrmions for a short period of time and then the skyrmion depins (for instance blue meaning probabilities of 20%). Finally, the very dark blue background corresponds to the area of low probability density where skyrmions occur only in few or single frames throughout the measurement (for further details about the analysis, see Supplementary Note 1 with Supplementary Fig. 1).

Figure 1b shows the energy landscape for the sample area corresponding to the probability distribution from Fig. 1a. We see a non-flat surface with distinct valleys connected to pinning sites and peaks for the positions with few observations. Thus, our method allows us to quantitatively ascertain the full energy

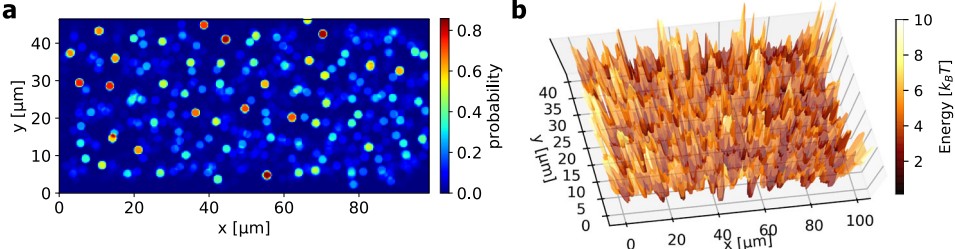

**Fig. 1 Skyrmion energy landscape. a** Occurrences of skyrmions in a 45 × 100 μm² film are measured at 312.5 K and −88 μT. The color scale represents the probability of pixels to be covered by skyrmions throughout the whole measured time. For example, at red positions, skyrmions were present around 80 % of the whole measurement time whereas the dark blue areas are rarely visited. The skyrmions explore the full space of the sample, only few single pixels are never covered by skyrmions. **b** Relative energy landscape surface in $k_BT$ determined from the probability distribution at 312.5 K in (**a**).

landscape of a sample and these results can then be used to gauge the applicability for nonconventional computing schemes.

**Single skyrmion pinning**. Having identified the presence of individual pinning sites, the next step is to investigate the underlying pinning mechanism for single skyrmion pinning. To realize this, we focus on a small confinement structure in a Ta(5)/$Co_{20}Fe_{60}B_{20}$(0.9)/Ta(0.08)/MgO(2) stack (thicknesses in nm, details in the methods section) presenting a non-uniform probability density. We study a single skyrmion in this area, which allows for observing thermal skyrmion dynamics governed only by the energy landscape and excluding any skyrmion-skyrmion interaction effects.

When we analyze the skyrmion probability density map, we find surprisingly that skyrmions of different sizes pin in very different positions. We vary the size (core area) of the skyrmion by up to a factor two using different applied fields at a certain temperature[26,44]. We start our analysis by the conventional effective potential rigid quasi-particle model investigating the distribution of the positions of the effective center of mass of the skyrmion. All observed skyrmion center occurrences at out-of-plane field values between −43 and −31 μT are shown in a scatter plot in Fig. 2a. The color indicates the size of the skyrmions located at the observed coordinates. The color opacity reflects the probability of finding a skyrmion there. In the measured field range, the mean skyrmion radius varies from 1.27 μm at −43 μT up to 1.68 μm at −31 μT. For comparison, those skyrmion sizes are indicated schematically below the scatter plot in Fig. 2a and in the color code established there. The histogram on the top shows the size distribution from all measurements.

As a key finding we see, that at different magnetic field values we find different pinning sites at which skyrmions are observed to be predominantly positioned. And at each of the pinning sites, the detected skyrmions are of a specific size. Thus, the pinning position of the skyrmion center and the skyrmion size are strongly correlated. To corroborate this concept, we study the probability distributions for skyrmions at distinct field values. Figure 2b–e show histograms of the occurring skyrmion center coordinates for external fields between −39 and −33 μT. Also, the size distribution of the skyrmions observed there is depicted with the color code as used before. We find a strong size dependence of the positions where the skyrmion center is pinned. For instance, when we vary the applied magnetic out-of-plane (OOP) field by only 2 μT corresponding to a skyrmion size change by 5–8 %, the obtained probability density distribution of the skyrmion center position varies drastically. This means that by varying the field and thus the size, we can switch on and off certain pinning sites and thus tune the pinning on-the-fly.

Skyrmions observed for −39 μT and below are relatively small with an average size 1.33 μm and mainly observed to be pinned at

pinning site 1 as depicted in Fig. 2d. Increasing the skyrmion size, Fig. 2c, d show that various other pinning sites become prominent. For the large skyrmions with an average radius of 1.63 μm at fields of −33 μT and for higher fields yielding larger skyrmions, only pinning site 3 is effective and skyrmions depin from all other pinning sites. Throughout all measurements, the skyrmions exhibit thermal dynamics and by varying the field we can pin them but also then depin them from all pinning sites. This means that the pinning is flexible and that we can move skyrmions between different positions on-the-fly by tuning their size: once a skyrmion becomes larger or smaller than the characteristic size of a pinning site, this site no longer exhibits pinning behavior. Since there is no pinning site at which the skyrmions are permanently trapped and cannot be depinned from, there is no universal deep minimum energy position with strong attraction and permanent trapping. Hence, we can flexibly tune the position where skyrmions are pinned.

For all investigated skyrmion diameters we identify pinning sites so that the energy landscape is found to be clearly nonflat with the probability density of finding skyrmions being strongly nonhomogeneous. In particular, we see that the distances between the observed pinning sites are smaller or comparable to the skyrmion radius. Hence, the area of a skyrmion core may cover several of the sites 1–4, which correspond to pinning sites for the skyrmion centers of mass. We notice however that for such large skyrmions of micrometer size, the core is homogeneously magnetized. Thus, the translation of the homogeneous core corresponds to a zero mode[45] and cannot influence the pinning site. In contrast, the skyrmion boundary (SB) corresponds to a large gradient in magnetization and to a nonzero energy density being sensitive to the energy landscape as especially the high energy of the boundaries favors being compensated by low potential regions[45]. Figure 3 provides a schematic of how skyrmions are pinned by the SB in the non-flat potential and how the energetically most favorable position changes with skyrmion size.

Due to a fixed SB, skyrmions of characteristic size still have specific center positions. The corresponding pinning of the center as presented in Fig. 2 and suggested in the conventional picture of rigid particles or effective potentials acting on the centers of skyrmions[31–38] is therefore an effective approach. For the large skyrmions investigated in this paper, the energy landscape is responsible for a more complex pinning behavior.

**Shape of pinned skyrmions**. We notice that pinning of the skyrmion governed by the boundary implies not only a size dependence of the skyrmion but also a shape dependence. To understand the mechanism of the pinning, we next study the detailed shape of the skyrmions when they are pinned at different pinning sites. For this we carry out high resolution Kerr

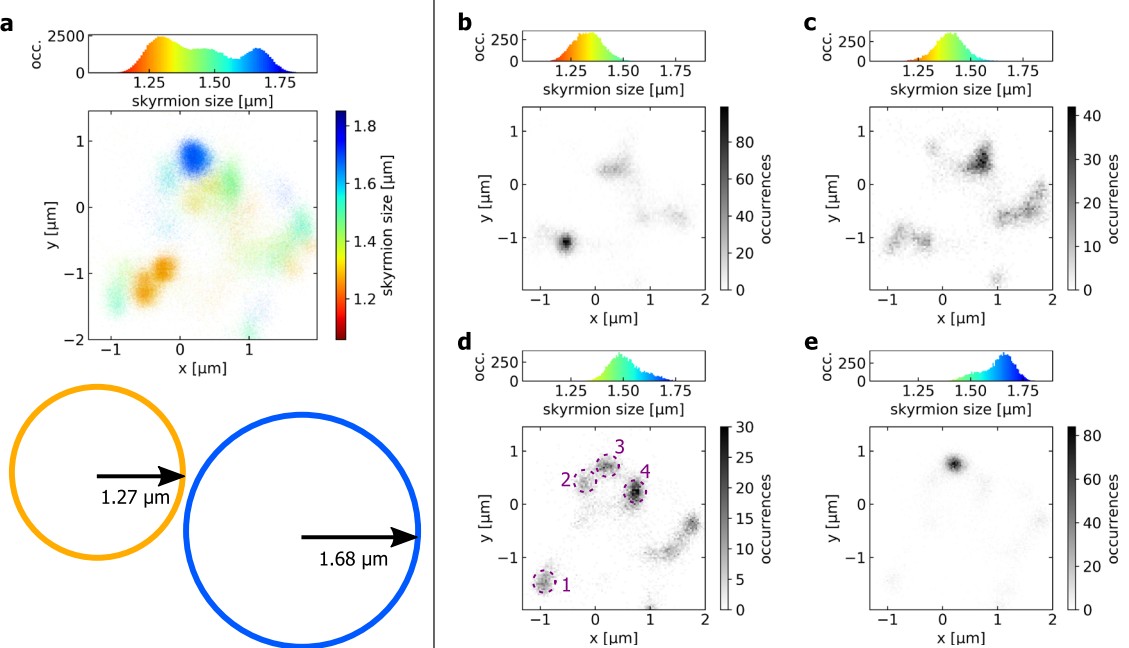

**Fig. 2 Skyrmion size dependence of the pinning. a** Skyrmion occurrences in all measurements at fields between −43 and −31 μT with an average skyrmion size between 1.27 and 1.68 μm with standard deviations of about 5%. Every detected skyrmion is depicted by a scatter plot point at the coordinate of observation. The colors represent the skyrmion size. The color intensity indicates the probability density of finding a skyrmion at a certain position. The top histogram shows the size distribution of all observed skyrmions. The colored circles on the bottom indicate schematically the size relation of the skyrmions among each other and the coordinate frame. **b–e** Histograms of the occurring skyrmion center positions in the same sample area at external fields of (**b**) −39 μT, (**c**) −37 μT, (**d**) −35 μT, and (**e**) −33 μT. The greyscale denotes the number of skyrmion center occurrences at a position. Even for small field changes of 2 μT corresponding to size changes of 5–8% in region depicted by (**b–e**), the skyrmion distribution varies drastically and skyrmions are pinned at different positions indicating size-dependent pinning sites. In (**d**), predominant pinning sites are highlighted by the dashed purple circles and labelled with integers 1–4.

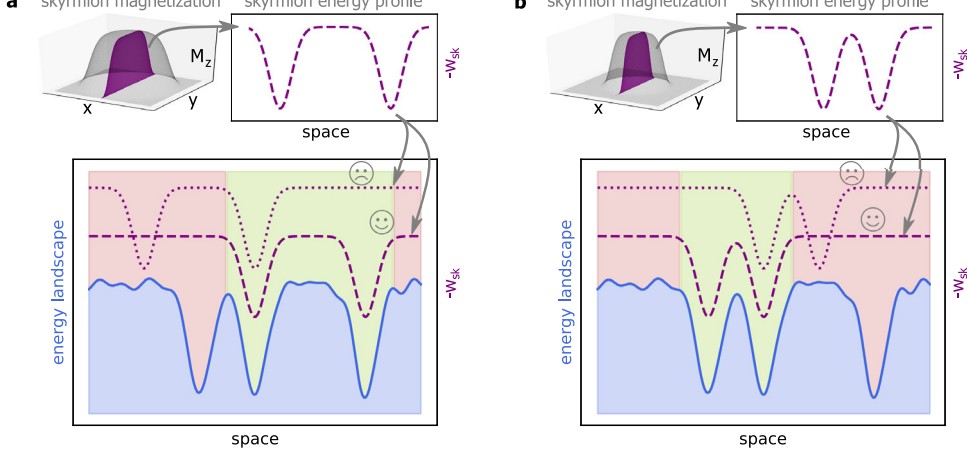

**Fig. 3 Schematics of skyrmion pinning in a nonflat energy landscape. a** The 3D surface schematically shows the magnetization of a skyrmion. The dashed line plot besides shows the negative energy density -$w_{sk}$ of this skyrmion along a profile as shown by the purple plane. It indicates that the (exchange-) energy density is most important in the domain walls delineating the skyrmion. The bottom part shows an exemplary schematic of a non-flat energy landscape in blue exhibiting three distinct minima. The peaks in energy density of the specific skyrmion match two of the valleys which leads to a favorable state of low energy in the region shaded in green as indicated by the dashed purple line. Other states as in the case of the dotted purple line yield a higher energy and are therefore unfavorable to be pinned at this position as indicated by the red-shaded region. **b** shows a similar scenario for a smaller skyrmion. Here, a minimal energy is achieved at a different position where the skyrmion energy density best matches the energy landscape (dashed purple line in the favorable green region). Hence, the position which was favorable in (**a**) is now in the unfavorable red region.

microscopy. The central coordinate frame in Fig. 4a shows the skyrmion shapes by plotting the positions of the domains walls that constitute the SB for skyrmions at pinning sites 1–4. The SB is defined as the position where the Kerr intensity intersects the mean of the values belonging to the skyrmion core and the

ferromagnetic background. The average Kerr contrast intensity of skyrmions is depicted in the surrounding plots for every pinning site. We see that although the corresponding skyrmion center coordinates deviate, their boundaries coincide in significant parts of the boundary length. To demonstrate the relevance of this

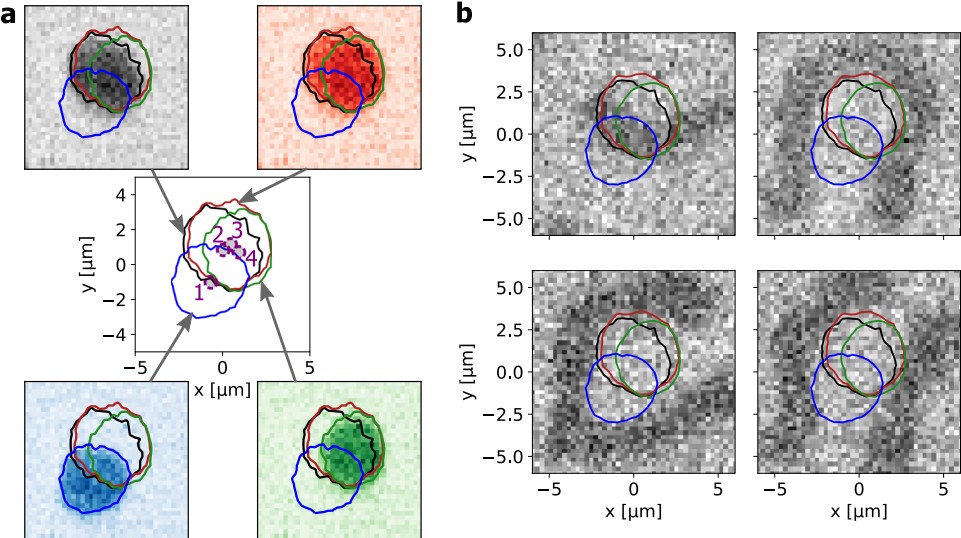

**Fig. 4 Pinning of skyrmion boundaries. a** The central coordinate frame shows the observed skyrmion boundary positions of the skyrmions at −35 μT at pinning sites 1 (in blue), 2 (black), 3 (red) and 4 (green) as introduced in Fig. 2d. The positions of the corresponding skyrmion center coordinates are visualized by the dashed purple circles, which are additionally filled in grey for better visibility. The surrounding plots depict the average intensity for each pinning site with a color scale matching to the boundary color. **b** Kerr microscopy images of arbitrary stripe observations in the same sample area. Black corresponds to magnetization pointing out of the plane while white denotes magnetization pointing into the plane. The previously determined skyrmion boundaries are plotted additionally for comparison. The paths of the skyrmion boundaries fit the domain boundary position of the stripes in significant parts.

effect, we present an additional example for differently sized skyrmions pinned at the same position is presented in Supplementary Note 2 with Supplementary Fig. 2. Note that for such large skyrmions, translation and deformation of the SB in a flat potential require very little energy[45]. Thus, recurring SB positions that coincide between different configurations provide strong evidence that pinning has to be present at the SB. However, the distance between the clearly distinct clusters of pinning sites 1 and sites 2–4 is of the order of or larger than the skyrmion radius and suggests that the difference between the pinning sites cannot be associated solely with a change in shape but the size is crucial. The equilibrium size of a skyrmion is thus governed by the applied magnetic field and temperature, based on the fundamental magnetic material properties[14,44,46].

To corroborate this pinning mechanism further, we explore pinning of stripe domains: the pinning mechanism should also be visible when studying domain walls in the stripe domain phase, rather than in the skyrmion phase. By tuning the OOP field and the nucleation process we obtain stripe domains as shown in Fig. 4b in the same confined sample as used for the skyrmion observation (single frame examples of stripe domains originating from different nucleation events). The domain walls of the stripe domains match the contours where the skyrmion domain wall boundaries were observed previously.

Having identified the domain walls at the skyrmion boundary as decisive for the pinning, we finally demonstrate how such a mechanism can arise. We use micromagnetic simulations with an arrangement of regions, which pin the domain wall. Exemplarily, the anisotropy is reduced in these regions as it is the magnetic property most sensitive to structural variations in the film and in particular the interfaces. As shown in Fig. 5, we can reproduce the observed pinning behavior by reducing the perpendicular magnetic anisotropy in three regions which are indicated as red boxes. Similar as in the experiment, the trajectory of a single skyrmion is tracked inside a small confinement and the skyrmion center positions are different when varying the skyrmion size due to the skyrmions being pinned with their boundary. Figure 5a, b shows histograms of the occurring skyrmion center coordinates

for both skyrmion sizes. A strong dependence of the center accumulation positions on the skyrmion size is observed, qualitatively supporting the experimental findings. Moreover, distinct skyrmion center accumulation points are observed although the underlying mechanism is pinning of the skyrmion boundary and not of the skyrmion center. The size of the smaller skyrmion is 5.7 nm meaning that its boundary can overlap only with two of the three pinning areas whereas the boundary of the larger skyrmion can overlap all three areas with a mean size of 10.3 nm. The occurring skyrmion sizes are shown in a histogram for Fig. 5a, b and furthermore, the mean sizes with respect to the coordinate frames used are depicted in Fig. 5c. Note that the SB usually makes up a much larger proportion of the skyrmion for nanometer-sized skyrmions than for the micro-scale skyrmions used experimentally. Given that the domain walls are the most highly energetic part of the skyrmion because of the rapid variation in the magnetization direction on a nm scale as compared to a homogeneously magnetized core of μm extension in the experiment, one can understand why variations in the magnetic properties most strongly influence the pinning of the skyrmion boundary domain walls. Further details on how the skyrmions of different size arrange in the simulation setup are found in Supplementary Note 3 with Supplementary Fig. 3.

## Discussion

Our results show that thermal skyrmion dynamics is a powerful method to ascertain the energy landscape of a sample. As a key finding, we demonstrate that the skyrmions explore by thermal dynamics nearly the full area of a state-of-the-art thin-film device and exhibit significant pinning at certain locations with a particular distribution. This highlights that the energy landscape contains distinct minima but virtually no highly repulsive maxima. Since the pinning positions are featured at different temperature and out-of-plane field values, we deduce the behavior to be based on local variations in the materials parameters[26,47,48] originating from the thin film growth process. As shown in our previous work on such samples, the observed pinning sites in this

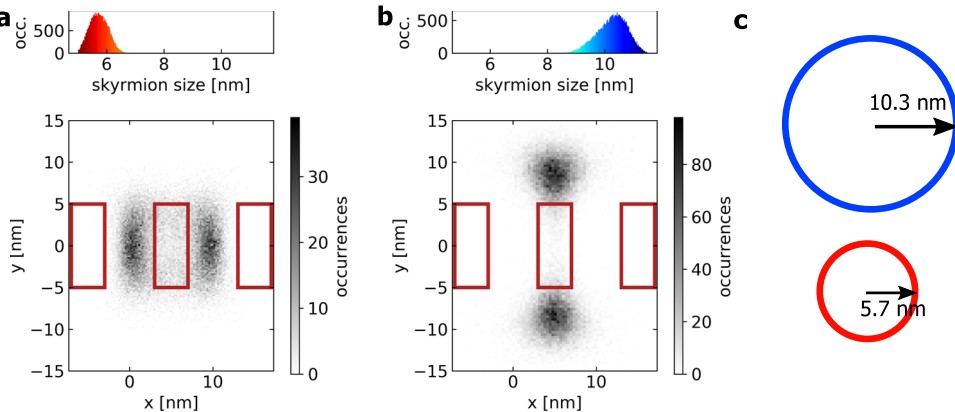

**Fig. 5 Skyrmion boundary pinning simulation.** Histograms of occurring skyrmion center positions for external out-of-plane fields of (**a**) 0.15 T and (**b**) 0.05 T based on micromagnetic simulations. The greyscale represents the number of skyrmion center occurrences at a position. Red boxes indicate the area where the magnetic anisotropy is reduced. The top histograms show the skyrmion size distribution. **c** Size comparison of the average sizes from (**a**) and (**b**). The scaling is identical for (**a–c**).

sample are weak enough so that driving skyrmions by spin-orbit torques is able to overcome the pinning and move skyrmions[14]. So the combination of quantified pinning sites and a possible skyrmion manipulation bodes well for applications in non-conventional computing since for instance reservoir computing needs pinning to ensure the reservoir's stability. A quantitative description of the pinning strengths and locations, as obtained in this work, gives access to the reservoir's performance, in particular regarding the reservoir's complexity, memory, and its nonlinearity[22,49].

We find that the pinning sites, at which skyrmion centers are located, vary drastically with the skyrmion size. In particular since the skyrmion size can be directly tuned by the applied external field at a fixed temperature[14,26,44,46], this observation allows for an experimental control of the skyrmion pinning sites that goes beyond the previously assumed static pinning energy landscape. In particular, we can actively manipulate the efficiency of the pinning sites by varying the skyrmion diameter and thus tune the effectiveness of single pinning sites: once the size of the skyrmion is incommensurate with the pinning site, the pinning site is effectively shut off. This allows for an unprecedented engineering of the skyrmion energy landscape on-the-fly during device operation, which might prove useful for a possible application with tunable pinning as theoretically suggested[23].

By magnetic microscopy, we find that the spacing between the distinctly observable positions where center positions of differently sized skyrmions accumulate, is sometimes smaller than the skyrmion radius. The experimental evidence shows that the observed effects cannot arise due to the pinning of the skyrmion core, which is uniform over an area exceeding the distances of the pinning sites. Instead, the skyrmion pinning originates from a mechanism that pins the domain walls at the skyrmion boundary (SB). From the direct observation of SB position, we conclude that the observed pinning sites arise due to the SB being pinned along certain complex contours. We furthermore note that since the obtained distinct regions where the skyrmion tends to be pinned are much smaller than the homogeneous core of the skyrmion, the skyrmion is not pinned at a point-like grain but rather by a distribution of points within a region of at least the size of the skyrmion. Due to characteristic occurring sizes, still distinct skyrmion center positions arise but are not very meaningful measures for the present pinning. In general, pinning yields a spatially varying additional energy contribution for the equilibrium state compared to the case without pinning and thus impacts the skyrmion shape. At a constant applied field, a

distribution of skyrmion sizes is observed and in particular, it becomes clear that the detailed energy landscape is responsible for the size and shape of a skyrmions occurring at a specific position. Finally, we can conclude that this mechanism is decisive for the position that the skyrmion is pinned at and determines the path of the contour along which the SB is positioned. From the shape-determining SB pinning mechanism, we especially deduce that conventional rigid particle descriptions or effective potentials are not sufficient to portray the actual complex pinning mechanism.

Since the thermal excitation is competing with the pinning energy on the same energy scale, the skyrmion is able to move in the sample region in the non-flat energy landscape. Moreover, it can undergo size changes of on average 5–10 %. Increasing its size, a skyrmion can match two different pinning features with different sections of its boundary. For example, while skyrmions at sites 2 and 4 show approximately the same size of around 1.5 μm, site 3 yields characteristic values above 1.7 μm. Thereby, the energy difference arising from the size variation occurring even at a fixed field can be explained by a compensation due to the pinning. In particular, both states featuring different skyrmion size and pinning are accessible by the thermal fluctuations present. Furthermore, we clearly observe that not only the size is determined by the pinning, but also the shape. Skyrmions at site 2 are elongated with the average eccentricity of 0.13. The resulting shape can again be well explained by the SB being pinned along a contour as set by the grain level such that an enhanced coincidence of SB and pinning feature may be energetically more favorable than the equilibrium shape in a flat potential. SB pinning introduced in micromagnetic simulations reproduces the size-dependent skyrmion center positions, which cannot arise from previously often assumed pinning due to the skyrmion core. The boundary pinning concept is further corroborated by the analysis of randomly generated static configurations of stripe domains which are shown to share the same boundary positions as the investigated skyrmions.

Since antiskyrmions[50] are also surrounded by domain walls that delineate the spin structure[45], the boundary-induced pinning effect should occur analogously to the skyrmion case studied here. In the case of antiferromagnetic skyrmion-like textures[51,52] however, the specific influence of the magnetic parameters that are unique to (synthetic) antiferromagnets, such as local variations of the antiferromagnetic exchange between the sub-lattices (respectively the RKKY between the two layers) will lead to additional effects[53,54].

In conclusion, pinning is a key feature relevant for many skyrmion-based spintronic devices. It allows to control many aspects of skyrmion dynamics such as nucleation, direction of motion, and speed. In racetracks memory concepts, for example, pinning allows to control the inter distance between skyrmions[12,55–57]. Pinning is even more essential for devices, which exploit skyrmion diffusion as in neuromorphic[58–60], probabilistic[14,61], reservoir[22,49] or Brownian computing[16,62]. Thus, pinning effects must not only be taken into account when realizing applications but certain pinning is actually required for the operation of some non-conventional computing devices and so understanding the pinning is a key requirement to ensure functionality. Whereas in some cases, the particle-like approach with the skyrmion being pinned in the center of site works, we find that in other cases it is the SB, which is pinned – depending on the skyrmion size and stiffness, as well as the energy landscape. Thus, with our findings, the pinning effect can even be exploited in new kinds of applications. For instance, a parameter variation as in terms of the anisotropy or DMI that can be tuned by electric fields or simply by varying the applied external magnetic field as shown here, we can switch on or off pinning at certain sites. This provides a flexible means to control skyrmion dynamics for potential new applications. And by tuning the pinning efficiency, applications such as reservoir computing or other non-conventional computing schemes can be enabled.

## Methods

**Sample characterization**. Multilayer stacks of Ta(5)/Co$_{20}$Fe$_{60}$B$_{20}$(0.9–1)/Ta(0.08)/MgO(2) as used in previous studies[14,63] are investigated. The values in parentheses indicate the layer thickness in nanometers. After deposition, the stacks are annealed at 250 °C at vacuum pressure to obtain perpendicular magnetic anisotropy.

For the energy landscape analysis, the skyrmions are observed within a confinement structure of $60 \times 120 \, \mu m^2$. However, the extension of the device is large enough to be treated as continuous film with respect to used skyrmion sizes, the governing diffusion constant and the observation time scales[14].

For the detailed study of magnetic pinning at a certain position, a disc with 17 μm diameter as previously investigated for diffusion in confinement[63] is used to ensure a single skyrmion to be present in a specific region.

The stacks were deposited by DC magnetron sputtering with a Singulus Rotaris sputtering tool using a base pressure of $3 \times 10^{-8}$ mbar. With the mentioned tool, the layer thickness can be controlled precisely with a reproducible accuracy of less than 0.01 nm. The stacks used are comparable in terms of the temperature and magnetic field region where skyrmions occur as well as in the obtained skyrmion sizes showing good reproducibility.

All confinement structures were patterned by electron beam lithography (EBL) and Ar ion etching. The samples exhibit perpendicular magnetic anisotropy (PMA) and we confirmed that they host magnetic structures of non-trivial topology demonstrating skyrmion character[14,63].

**Measurement setup**. A commercial *evico magnetics GmbH* Kerr microscope including an in-plane (IP) field coil is used in combination with an electromagnetic coil for out-of-plane (OOP) field control which was custom-made at the University of Mainz. For this small coil without magnetic core, a precise current control is used to realize OOP fields of the order of microtesla and with sub-μT precision. The field values are calibrated with a sensitive Hall probe, corrected by the hysteresis offset and reproduced in several measurement series. The magnetic structures are made visible exploiting the polar magneto-optical Kerr effect (MOKE) using a time resolution of 62.5 ms corresponding to 16 frames per second. To enable heating, the sample is placed on a Peltier element and a Pt100 sensor is attached to the Peltier element right next to the sample for temperature control. The temperature stability was ensured to be within 0.1 K.

The samples are heated by a Peltier element to temperatures between 300 and 350 K in order to get stable skyrmion phases exhibiting thermal motion[14].

**Investigated skyrmion systems**. The skyrmions are nucleated at measurement temperature and OOP field by additionally applying an IP field sweep. During a sweep, a saturating IP field is rapidly switched off to relax spins into an equilibrium state. We are able to control the number of skyrmions in the geometry by the field strength to obtain desired skyrmion densities[63].

To quantify and investigate the occurrence of pinning in the continuous sample, an array of many skyrmions is nucleated. The high number of skyrmions helps to acquire statistically relevant skyrmion occurrences throughout the whole region within the observation time. However, the skyrmion density is kept sufficiently low

with the skyrmions appearing at distances of several skyrmion diameters so that a significant influence of repulsive skyrmion-skyrmion interactions on the thermal motion can be ruled out[14]. To obtain the necessary statistics to quantify the strength of single pinning sites but also to qualitatively observe where pinning sites occur on the sample, 20 videos of 9600 frames (ten minutes length) each with re-nucleated skyrmions are recorded. The re-nucleations thereby enable the skyrmions to occur at new positions each time. Especially, we ensure that skyrmion nucleation does not occur directly at pinning sites so that skyrmions exhibit significant thermal dynamics before they might eventually become trapped at a strong pinning site and thus the full space of the sample is explored and not limited by repulsive skyrmion-skyrmion interaction[14].

To study the skyrmion dynamics near pinning effects in a thin film, we nucleate a single skyrmion in a disc with 17 μm diameter. In this confined geometry, the skyrmion can explore the energy landscape but cannot escape from the disc. For the analysis of the experiment, we have selected a sample where we find that the skyrmion positions are not homogeneously distributed but influenced by pinning but there is no pinning site, which permanently traps the skyrmion. This furthermore enables to study the skyrmion dynamics even for differently sized skyrmions. The skyrmion was nucleated and recorded for ten minutes (9600 frames) for each skyrmion size related to a certain value of the OOP field value. As the skyrmion is continuously moving between different positions and thus never permanently trapped at a pinning site, multiple nucleation events are not required in this case.

The skyrmion size can directly be tuned by the applied magnetic out-of-plane field[14,46] and temperature[14,44]. With our measurement at constant temperature, we can thus tune the average skyrmion size by the external field value[14,46]. We do not find that the skyrmion is ever leaving the disc or is annihilated inside the disc during the measurements for the field and temperature ranges studied.

**Skyrmion imaging and tracking**. The videos are acquired with the CCD camera of the microscope, the *trackpy*[64] package is used to preprocess the frame images and detect the skyrmions. The preprocessing includes both background gradient compensation and noise filtering. The detection is then performed by fitting two-dimensional Gaussian kernels for localized intensities exceeding certain thresholds in terms of intensity and size. The optical spatial resolution of the MOKE microscope (≈300 nm) is better than the skyrmion size but domain boundaries cannot be resolved. Changes in the magnetization lead to a significant intensity edge and the dimensions of the skyrmions can be determined with subpixel accuracy. The validity of those obtained quantities well below the single-pixel range was ensured by a wide variation of the tracking parameters and comparison of intensities of single frames as well as averaged over frames yielding quantitatively identical results. Therefore, a spatial resolution well below 100 nm is reached for the detection of the skyrmion center of mass positions. As the skyrmion size, the gyration radius of the fitted intensity profile is established by the tracking algorithm[64]. Note that this size is smaller than but still a measure for the actual extent of the magnetic texture.

The video contrast is enhanced by the microscope performing background subtraction. Since we have access to both unprocessed images and used background images, sample drifts occurring can be detected at the confinement edge and quantified with precision well below the single-pixel regime. In the analysis, the drift can thus be compensated. Furthermore, comparing the position of the confinement edges for every measurement makes it possible to establish a reference coordinate frame. All detected skyrmion positions are expressed in this coordinate system. There, the observations are compared with all the conducted measurements.

A further important part of the analysis is the detection of skyrmion boundary (domain wall) positions. These cannot be obtained directly from single-frame images since the magnetic contrast in images from the Kerr microscope is too low. The skyrmions observed here are of micrometer size and their center of mass can therefore be detected and located precisely since they cause an average difference in the intensity level over all the several pixels that they cover. However, the domain walls present at the skyrmion boundaries are of widths in the subpixel regime[14,63]. Therefore, the intensity edge originating from the transition between the skyrmion core and the ferromagnetic background would be expected to occur within one pixel but the present noise prohibits even a skyrmion boundary (SB) detection with pixel accuracy. Nevertheless, the SB can be determined accurately when averaging over similar frames due to gained statistics regarding the position and noise being averaged out. Therefore, the individual frames are assembled in groups such that each group contains skyrmions located at one specific position. The intensities of those frames are then averaged within every group to average out noise. Edge-preserving bilateral filtering is used to further reduce the noise present despite the frame averaging. The position of the SB per pinning site is then obtained as the contour where the filtered intensity profile intersects the mean value between core and ferromagnetic background level. The group elements are selected by skyrmion center coordinates corresponding to the centers of peaks in the probability density distribution. Due to the selection by a small range of center coordinates, we consider only skyrmions with negligible variations and hence ensure that the SB position is preserved in the averaging. Still, at least 350 corresponding to 3.7% of the total observations are considered per pinning site and hence allow for an adequate determination of the SB position.

**Micromagnetic simulation setup**. To understand and support the experimental findings, nano-scale micromagnetic simulations are performed using the *mumax3* simulation package[14,65–67]. Similar to the experimental study of the skyrmion inside a small confinement, a single skyrmion is simulated inside a square box with open boundary conditions. The side length is chosen to be about four times the size of the largest simulated skyrmion such that effects of the sample edge are negligible in the central region of the sample. There, different domain wall pinning regions are exemplarily created by locally lowering the effective PMA strength $K_{eff}$ to 20%. The skyrmion size is tuned via an external OOP magnetic field. The system evolves under the finite temperature Landau-Lifschitz-Gilbert equation for 60 µs and is sampled every 1 ns. As in the experimental setup, the *trackpy* package is used for skyrmion detection based on greyscale images of the OOP magnetization[64].

Typical nano-scale sample parameters were employed for a $128 \times 128 \times 1\,nm^3$ sample[68–70]. A cell size of $1 \times 1 \times 1\,nm^3$ is used and the sample has a saturation magnetization of $M_s = 1\,MA/m$, exchange stiffness of $A_{ex} = 15\,pJ/m$ and Gilbert damping of $\alpha = 0.01$. The demagnetization energy is accounted for by employing an effective PMA strength of $K_{eff} = K_u - 0.5 \cdot \mu_0 M_s^2$ with $K_u = 1.1\,MJ/m^3$, which is equivalent to considering an infinite thin film and explicitly calculating the demagnetization field in the case of a homogeneous magnetization[68,71]. In the presence of skyrmion-type magnetic spin structures, this formalism is however a well-established approximation for performance reasons, while furthermore not leading to boundary effects due to stray fields[68,71]. The interfacial DMI strength is chosen to $D = 0.95 \cdot D_{crit}$ with $D_{crit} = 4\sqrt{A_{ex} K_{eff}}/\pi$ and skyrmions are stabilized by applying external OOP magnetic fields of 0.05 T and 0.15 T for large and small skyrmions, respectively. The pinning areas have a size of $4 \times 10\,nm^2$ and the temperature is set to 100 K.

## Data availability

The data that support the findings of this study are available from the corresponding author upon reasonable request.

## Code availability

The codes associated with the evaluation of experimental data and the simulation presented in this paper are available from the corresponding author upon reasonable request.

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

## Acknowledgements

The authors acknowledge funding from TopDyn, SFB TRR 146 (project #233630050), SFB TRR 173 Spin+X (project A01 #268565370 and project B12 #268565370). The work was additionally funded by the Deutsche Forschungsgemeinschaft (DFG, German Research Foundation) Project No. 403502522 (SPP 2137 Skyrmionics) as well as the Emmy Noether project #320163632. This project has received funding from the European Research Council (ERC) under the European Union's Horizon 2020 research and innovation programme under grant agreement No. 863155 (s-Nebula) and No. 856538 (ERC-SyG 3D MAGIC). Open access funding is enabled and organized by Projekt DEAL. J.Z. acknowledges the support of Charles University grant PRIMUS/20/SCI/018.

## Author contributions

M.K., P.V., and J.Z. supervised the study. N.K., B.S., and M.V. fabricated and characterized the multilayer samples. R.G. prepared the measurement setup, conducted the experiments using the Kerr microscope and evaluated the experimental data with the help of J.Z.; M.B. performed the micromagnetic simulation with guidance of D.R. and evaluated the simulation data. R.G prepared the manuscript with the help of M.K., P.V., D.R., K.E., J.Z., M.B., and T.D.; all authors commented on the manuscript.

## Funding

## Competing interests

The authors declare no competing interests.
