## [Peer Review File · Nature Communications]

Reviewers' Comments:

Reviewer #1:

Remarks to the Author:

Skyrmion pinning energetic in thin film systems

R. Gruber et al

Skyrmions are magnetic topological objects that have attracted considerable attention for both basic science and possible applications. There has been a large amount of work looking at how to stabilize skyrmions. In order to understand more fully skyrmion dynamics and use them in applications, it is necessary also to understand skyrmion pinning effects and this is a growing area. Although there have been many assumptions made theoretically about pinning, the actual microscopics of skyrmion pinning in real systems is not very well understood. The point of this work is to better understand the energetics of skyrmion pinning. Here the authors show the shape and edge of the skyrmion as opposed to the core. The authors show a complex skyrmion energy landscape. The authors demonstrate that skyrmion size will also show strong effects. The work is important and should be useful for both experimentalists and theoreticians studying skyrmions and could lead new ideas for devices. The authors also have a very nice discussion section of the overall results. I think the results could be very important but could be better explained, particularly Fig. 2 which could be updated by the addition of a schematic.

Specific comments for the authors to address:

(1) One of the results is the skyrmion size effect of the pinning where the authors argue that changing the size of the skyrmion with a field can be used effectively to switch on and off the pinning.

The writing was a bit awkward to me. Perhaps the authors should include a very clear statement like "Once the size of the skyrmion is larger than the pinning site the pinning site is effectively shut off." I think that is what the point of that section is. They could also have a schematic of the effective pinning force versus skyrmion size and show that it drops off rapidly once the size is large enough. Figure 2 has some nice information but will be difficult for non-experts to really grasp; that is why I think a schematic or two could also help. Since this is a key point this needs to be explained more clearly. There should be room in the text for this.

Also, there is some current experimental and theoretical work in the literature that also suggests the pinning effectiveness will be affected strongly by the skyrmion size. In fact, the size dependence could be used to explain some of the results in the skyrmion Hall effect,

"Visualizing the strongly reshaped skyrmion Hall effect in multilayer wire devices",
A.K.C Tan, P. Ho, J. Lourembam, L. Huang, H.K. Tan, C. J. Olson Reichhardt, C. Reichhardt, and A. Soumyanarayanan
Nature Communications 12 4252 (2021).

Another work the authors could also comment on is

Zeissler, K. et al. Diameter-independent skyrmion Hall angle observed in chiral magnetic 455 multilayers. Nat. Commun. 11, 428 (2020)

as discussed in

"Plastic flow and the skyrmion Hall effect",
C. Reichhardt and C.J. Olson Reichhardt
Nature Commun 11 738 (2020).

These works found that the skyrmion Hall effect was independent of the size of the skyrmion due to the pinning. One possible explanation is that in the Zeissler experiments, the pinning disorder was on a much larger length scale than the skyrmions, which would also explain why the skyrmion

went along the same paths. In the work of A.K.C. Tan et al they discuss how the results could be explained better if there is a strong skyrmion size dependence on the pinning. This should be mentioned in the discussion.

The authors could also add some comments on the possible effects on the skyrmion Hall effect, since it could be possible that once the skyrmions are large enough the skyrmion Hall effect is no longer modified by the pinning, while for small skyrmions the pinning has a large effect on the skyrmion Hall effect.

(2) Since the authors only have three figures, I think it might also be nice to move one of the micromagnetic simulation results from the supplemental material to the main text.

The authors should add references on work on the studies of pinning in general to other particle like systems.

"Depinning and nonequilibrium dynamic phases of particle assemblies driven over random and ordered substrates: a review ",
Charles Reichhardt and C.J. Olson Reichhardt
Rep. Prog. Phys 80 026501 (2017).

(3) Along the lines of these discussions ,the authors could also mention how the pinning might be different for antiferromagnetic skyrmions or antiskyrmions. I understand this is beyond the scope of this work but they could add a few comments along this direction.

Reviewer #2:

Remarks to the Author:

In this work, the authors focus on skyrmion pinning. Skyrmion pinning is the most disgusting obstacle in skyrmion science as compared to its beautiful topological nature. We have not talked about it positively, although we have to cope with it and eventually control it in device applications. As the authors point out, skyrmion pinning is more serious for non-conventional computing devices than conventional ones. We know that uncontrolled pinning causes unnecessary energy consumption, even for conventional devices. The theme of the work is timely and worthy of publication.

They worked on three subjects.

1) Determination of the potential energy for a single skyrmion:

They obtained the potential energy, the energy landscape in their terminology, from the skyrmion probability density.

2) Skyrmion size dependence of pinning:

Pinning depends on the size of the skyrmions. Indeed, it is an exciting finding.

3) Mechanism of skyrmion boundary pinning.

They claim that the skyrmion boundaries play a decisive role in pinning.

I have reviewed the manuscript of R. Gruber and co-authors. I enjoy this challenging manuscript but still am puzzled by some parts. The Authors should clear up the following questions before the manuscript can be published.

1)About skyrmion size dependence of the pinning:

In Fig.2, pinning strength is not proportional or monotonic to the size of skyrmions. Does it suggest there is a characteristic size of the pinning site?

2)About pinning of skyrmion boundaries:

The meaning of boundary pinning is not clear to me. In Fig.3, Skyrmions are simply captured in a low potential-energy region, which becomes distinct by the stripe pattern with a lower OOP field. The observed pinning sites (called 1~4) are attributed to the complex shape of the pinning region. How can one exclude this simple argument? Does it also explain the first puzzle of skyrmion size dependence?

Recently, Ishikawa et al. published a paper about not pinning but an artificial skyrmion cell that confines skyrmions. {Appl. Phys. Lett. 119, 072402 (2021)}
Ishikawa et al. observed somewhat similar to this paper.

3) About skyrmion boundary pinning simulation:

The results of the simulation lead to the conclusion of the paper. However, I believe that the employed simulation condition is not suitable. In simulations with open boundary conditions, the demagnetization (dipole) fields are excited and make extra potential holes inside. Therefore, one has to prepare a more extensive simulation field to avoid the boundary effects. The effect was verified experimentally by [16] Jibiki et al.

In summary, I believe that extensive micromagnetic simulations are indispensable because the conclusions of this paper rely strongly on the simulation.

Reviewer #3:

Remarks to the Author:

The authors study the positions of $\sim 1\mu\text{m}$ -wide skyrmions in a CoFeB film that drift by thermal excitation. By examining the residence times, the authors can deduce a 2D map of skyrmion energy versus position, with the implied approximation of the skyrmion by a point particle. Examining the pinned skyrmion at a scale comparable to its diameter, the authors find that the location of the 'centre-of-mass' of the skyrmion depends on its diameter, which the authors change by applying a field. From this, the authors claim to observe pinning sites that are dependent on the skyrmion size. The authors also observe the shape of the pinned skyrmion, and observe that the borders of the different-sized skyrmions share a common area. From this the authors deduce that the assumption that pinning is determined by the skyrmion's centre is false, an assumption that the authors claim to be commonly accepted.

I agree with the authors that problem of skyrmion pinning is interesting, and the observation of skyrmion residence times is an interesting tool to study it. However, I have significant reservations regarding the second and main part of the work. The variation of the skyrmion centre-of-mass with skyrmion size is claimed to be a proof of a variation of pinning sites with skyrmion size. However, the studied skyrmions are larger than the separation between the supposedly different pinning sites. The phenomenon is more straightforwardly understood as a variation of the shape (and area) of the skyrmion with applied field, pinned at the same site — a conclusion that the authors seem also to share when presenting fig 3. The data of Figs. 2 and 3 do not show, in my opinion, that a "surprisingly strong skyrmion size dependency of the pinning". Instead, they show a small — relatively to skyrmion diameter — variation of the skyrmion shape. That claim would be supported by a figure in the scale of Fig. 1a for different skyrmion diameters.

Another reservation I have is regarding the claim that it is commonly-accepted that the skyrmion's centre is the critical region in the pinning process. No references are given to support that this is a commonly-accepted hypothesis. Indeed, a 'naïve' hypothesis of skyrmion pinning would state exactly the opposite: pinning is a variation of energy with position (dU/dx), therefore the skyrmion centre, where magnetisation is constant ($dM/dx=0$), should not directly influence pinning! It might be common in the literature to describe pinning sites using the skyrmion's centre-of-mass (as in Fig. 1), with the implied approximation of the skyrmion by a particle. However, the data reported in this manuscript does not invalidate this description for the length scales larger than the skyrmion diameter, which is its reasonably expected condition of validity.

I find that, although the experiments seem to have been carefully performed and analysed, some of the main claims of the work are not supported by the experimental observations. The study of skyrmion residence times (Fig 1) is interesting. The authors do show that the pinned skyrmion centre-of-mass varies with skyrmion size. However, I do not find that these conclusions are sufficient to warrant publication in Nature Communications.

POINT-BY-POINT REPLY TO REVIEWER COMMENTS

References are numbered identically as in the revised manuscript. An overview over major changes to the manuscript is provided at the end of this document.

Reviewer #1 (Remarks to the Author):

Skyrmion pinning energetic in thin film systems

R. Gruber et al

Skyrmions are magnetic topological objects that have attracted considerable attention for both basic science and possible applications. There has been a large amount of work looking at how to stabilize skyrmions. In order to understand more fully skyrmion dynamics and use them in applications, it is necessary also to understand skyrmion pinning effects and this is a growing area. Although there have been many assumptions made theoretically about pinning, the actual microscopics of skyrmion pinning in real systems is not very well understood. The point of this work is to better understand the energetics of skyrmion pinning. Here the authors show the shape and edge of the skyrmion as opposed to the core. The authors show a complex skyrmion energy landscape. The authors demonstrate that skyrmion size will also show strong effects. The work is important and should be useful for both experimentalists and theoreticians studying skyrmions and could lead new ideas for devices. The authors also have a very nice discussion section of the overall results. I think the results could be very important but could be better explained, particularly Fig. 2 which could be updated by the addition of a schematic.

Author's reply

We appreciate the referee's comments, and we thank him/her for considering our work as "very important" and "useful for both experimentalists and theoreticians". We are also grateful to the referee for the valuable comments that have helped us improve our manuscript further. Below, we provide a point-by-point reply to the comments raised by the referee.

Specific comments for the authors to address:

(1) One of the results is the skyrmion size effect of the pinning where the authors argue that changing the size of the skyrmion with a field can be used effectively to switch on and off the pinning.

The writing was a bit awkward to me. Perhaps the authors should include a very clear statement like "Once the size of the skyrmion is larger than the pinning site the pinning site is effectively shut off." I think that is what the point of that section is. They could also have a schematic of the effective pinning force versus skyrmion size and show that it drops off rapidly once the size is large enough. Figure 2 has some nice information but will be difficult for non-experts to really grasp; that is why I think a schematic or two could also help. Since this is a key point this needs to be explained more clearly. There should be room in the text for this.

Author's reply:

We thank the referee for these concrete comments and suggestions, which helped us improve our manuscript to realize a better explanation of our established size-dependent skyrmion boundary pinning concept to the reader. As the referee correctly points out, our investigations have revealed that depending on the landscape, certain areas of the landscape are only able to pin skyrmions of a specific size. The concept presented is very different from the conventional idea that is considered common

wisdom that there are pinning centers that pin rigid skyrmions. We have revised the text including a statement as suggested by the referee in both the results (“[O]nce a skyrmion becomes larger or smaller than the characteristic size of a pinning site, this site no longer exhibits pinning behavior”) and in the discussion section (“[...] once the size of the skyrmion is incommensurate with the pinning site, the pinning site is effectively shut off.”). We furthermore agree that a schematic is helpful to make the boundary pinning concept derived from Figure 2 better understandable to the reader. We therefore added an additional, schematic as a new figure (now Figure 3) to present this concept in a clearer way. We hope that this figure makes the concept clear and if the referee has any suggestions to improve it further, we are of course happy to hear about it.

Also, there is some current experimental and theoretical work in the literature that also suggests the pinning effectiveness will be affected strongly by the skyrmion size. In fact, the size dependence could be used to explain some of the results in the skyrmion Hall effect,

*"Visualizing the strongly reshaped skyrmion Hall effect in multilayer wire devices",
A.K.C Tan, P. Ho, J. Lourembam, L. Huang, H.K. Tan, C. J. Olson Reichhardt, C. Reichhardt,
and A. Soumyanarayanan
Nature Communications 12 4252 (2021).*

Another work the authors could also comment on is

Zeissler, K. et al. Diameter-independent skyrmion Hall angle observed in chiral magnetic 455 multilayers. Nat. Commun. 11, 428 (2020)

as discussed in

*"Plastic flow and the skyrmion Hall effect",
C. Reichhardt and C.J. Olson Reichhardt
Nature Commun 11 738 (2020).*

These works found that the skyrmion Hall effect was independent of the size of the skyrmion due to the pinning. One possible explanation is that in the Zeissler experiments, the pinning disorder was on a much larger length scale than the skyrmions, which would also explain why the skyrmion went along the same paths. In the work of A.K.C. Tan et al they discuss how the results could be explained better if there is a strong skyrmion size dependence on the pinning. This should be mentioned in the discussion.

The authors could also add some comments on the possible effects on the skyrmion Hall effect, since it could be possible that once the skyrmions are large enough the skyrmion Hall effect is no longer modified by the pinning, while for small skyrmions the pinning has a large effect on the skyrmion Hall effect.

Author's reply:

We thank the reviewer for suggesting these fitting references. We agree with the assessment of the referee that pinning can be important for the skyrmion Hall angle, and we added the suggested references with a discussion to the manuscript. Indeed, the pinning dependence on the size of the skyrmion may strongly impact the skyrmion motion and the skyrmion Hall angle in inhomogeneous samples. We have incorporated this comment to the revised manuscript. Indeed, it would be interesting to correlate the pinning with the skyrmion Hall angle and this might be an interesting topic for a future study but clearly goes beyond the scope of our current work.

(2) Since the authors only have three figures, I think it might also be nice to move one of the micromagnetic simulation results from the supplemental material to the main text.

Author's reply:

We are grateful for this comment and agree that moving the figure on the micromagnetic simulation results to the main text (now as Figure 5) is beneficial for explaining our findings and better highlighting the role of the simulations. To keep the text flow, we therefore also interchanged the simulation results with the observation of the stripe domains, which does not influence the message of our work since both investigations are found to stress our findings. Additionally, we added a new figure showing examples of simulated skyrmions to the supplementary material (Supplementary Note 3 with Supplementary Figure 3) to enable a better understanding of the simulation and the key results.

The authors should add references on work on the studies of pinning in general to other particle like systems.

*"Depinning and nonequilibrium dynamic phases of particle assemblies driven over random and ordered substrates: a review",
Charles Reichhardt and C.J. Olson Reichhardt
Rep. Prog. Phys 80 026501 (2017).*

Author's reply:

We appreciate the referee's comment, and we thank you for the suggestion of a very valuable reference. We therefore have additionally added the reference on general pinning studies in the beginning of the introduction, which puts the whole work in a context also beyond the field of skyrmions.

(3) Along the lines of these discussions, the authors could also mention how the pinning might be different for antiferromagnetic skyrmions or antiskyrmions. I understand this is beyond the scope of this work but they could add a few comments along this direction.

Author's reply:

We thank the reviewer for this comment. We agree that the detailed investigation of pinning for skyrmions in antiferromagnetic materials as well as for antiskyrmions is beyond the scope of this work. However, as this is an important topic, we included comments about this topic in the discussion section of the revised manuscript, which are outlined in the following. Our results suggest that the pinning is depending on the domain wall that forms the border of the large skyrmions. Since both large antiskyrmions and large skyrmions may be described by being surrounded by closed domain walls, the pinning mechanism is expected to be analogous to our observations. We note that the complex landscape that defines the pinning sites depends on the different magnetic parameters and inhomogeneities.

Concerning antiferromagnets, one has to take into account that the magnetic parameters play different roles for ferro- and antiferromagnetic skyrmions. So, the resulting pinning effects can change – depending on whether a specific pinning site is mediated by changes in the anisotropy or exchange interaction, for instance. For synthetic antiferromagnetic skyrmions, the effective energy landscape is furthermore governed by the coupling of the landscapes of the two layers, which is beyond the scope of this work, as well but we comment on this now in the revised and extended manuscript.

Reviewer #2 (Remarks to the Author):

In this work, the authors focus on skyrmion pinning. Skyrmion pinning is the most disgusting obstacle in skyrmion science as compared to its beautiful topological nature. We have not talked about it positively, although we have to cope with it and eventually control it in device applications. As the authors point out, skyrmion pinning is more serious for non-conventional computing devices than conventional ones. We know that uncontrolled pinning causes unnecessary energy consumption, even for conventional devices. The theme of the work is timely and worthy of publication.

They worked on three subjects.

1) Determination of the potential energy for a single skyrmion:

They obtained the potential energy, the energy landscape in their terminology, from the skyrmion probability density.

2) Skyrmion size dependence of pinning:

Pinning depends on the size of the skyrmions. Indeed, it is an exciting finding.

3) Mechanism of skyrmion boundary pinning.

They claim that the skyrmion boundaries play a decisive role in pinning.

I have reviewed the manuscript of R. Gruber and co-authors. I enjoy this challenging manuscript but still am puzzled by some parts. The Authors should clear up the following questions before the manuscript can be published.

Author's reply:

We thank the referee for acknowledging that our results are “timely and worthy of publication” and are grateful to the referee for his/her helpful comments on our manuscript which we address in detail below. In response to the comments, we have carried out additional work, which has strengthened our results and conclusions as detailed in the following.

1) About skyrmion size dependence of the pinning:

In Fig.2, pinning strength is not proportional or monotonic to the size of skyrmions. Does it suggest there is a characteristic size of the pinning site?

Author's reply:

We thank the referee for this comment raising the interesting point of the “size of the pinning site”. Indeed, one could effectively assign to a pinning site an effective size in the sense that skyrmions of a certain size are pinned at that pinning site. However, the effect that we observe is more subtle in that we consider skyrmions that are not perfectly round, and we have to go beyond the rigid object approximation to explain our results: We have introduced an additional paragraph and figure (Supplementary Note 2 with Supplementary Figure 2) to demonstrate explain further the mechanism of pinning by the energy landscape. In particular, the specific sizes pinned at certain positions are determined by the complex energy landscape rather than by a single pinning site. The energy landscape itself arises from the random distribution of pinning sites/inhomogeneities in the amorphous material and leads to pinning of the skyrmion boundaries that are the domain walls that delineate skyrmions. This is now clarified in the newly added schematic in Figure 3.

2) About pinning of skyrmion boundaries:

The meaning of boundary pinning is not clear to me. In Fig.3, Skyrmions are simply captured in a low potential-energy region, which becomes distinct by the stripe pattern with a lower OOP field. The observed pinning sites (called 1~4) are attributed to the complex shape of the pinning region. How can one exclude this simple argument? Does it also explain the first

puzzle of skyrmion size dependence?

Recently, Ishikawa et al. published a paper about not pinning but an artificial skyrmion cell that confines skyrmions. {Appl. Phys. Lett. 119, 072402 (2021)} Ishikawa et al. observed somewhat similar to this paper.

Author's reply:

We thank the referee for this comment and the interesting question whether our results could also be obtained in a cell of reduced energy. Indeed, we can exclude that such effects can explain our findings: We show in the revised Figure 4 (former Figure 3), that the boundaries of the skyrmions, even at different size and shape, tend to coincide. This reveals that the boundary plays an important role for the skyrmion pinning. We notice that these skyrmions are of μm size – hence in the bubble skyrmion limit– and smooth deformations of the boundary require very low energy compared to the energy of thermal fluctuations. Therefore, the fact that the boundaries coincide means that indeed there is a pinning at the boundary. We have included this remark to both the results and discussion section to emphasize the dependence of the skyrmion pinning on the complex, non-flat energy landscape given by the grain and inhomogeneities distribution. Furthermore, as stated above in reply 2.1, our findings are now further explained by our new schematic (Figure 2) and the additional Supplementary Note 2 with Supplementary Figure 2, which hopefully clarify the concept better.

3)About skyrmion boundary pinning simulation:

The results of the simulation lead to the conclusion of the paper. However, I believe that the employed simulation condition is not suitable. In simulations with open boundary conditions, the demagnetization (dipole) fields are excited and make extra potential holes inside. Therefore, one has to prepare a more extensive simulation field to avoid the boundary effects. The effect was verified experimentally by [16] Jibiki et al.

I summary, I believe that extensive micromagnetic simulations are indispensable because the conclusions of this paper rely strongly on the simulation.

Author's reply:

We thank the referee for the important comment. The referee is correct in that, in general, with open boundary conditions, the demagnetization (dipole) fields are excited and thereby the potential landscape is altered. However, the effect of demagnetization is included in our simulations by a (homogeneous) reduction of the perpendicular magnetic anisotropy $K_{eff} = K_u - 0.5 \cdot \mu_0 M_s^2$. Concerning stray fields, this approach is equivalent to considering an infinite film and explicitly calculating the demagnetization field in the case of a homogeneous magnetization [65,66]. In the presence of a skyrmion-type magnetic structure, this formalism is however a well-established approximation for performance reasons, while not leading to boundary effects due to stray fields in the thin film limit [65,66]. We have carefully considered the valid criticism about the wrong depiction in our original manuscript. In response to the referee's beneficial comment, we mention the above conditions of the stray field inclusion in the methods section on the micromagnetic simulation setup and added the respective references demonstrating that the performed simulations are indeed based on a valid approximation.

Moreover, we agree with the referee that the micromagnetic simulations constitute an important part of our conclusion. We point out that our conclusion regarding the skyrmion boundary pinning already follows from the observations depicted in Figures 2-4 (including the new schematic Figure 3). This is now also made more concrete in the new discussion of the boundary pinning concept based on the comments by reviewer #1. Furthermore, the role of the simulation as supporting our findings benefits from the rearrangement due to the interchange with the stripe domain investigation, which is a consequence of moving the simulation figure to the main text (as Figure 5) as suggested by review #1 as well. To avoid misconceptions, we correspondingly also adjusted the wording when referring to the

simulation. For instance, in the results section, we now state that we want to “demonstrate how such a [pinning] mechanism can arise” instead of that we “need to understand the [pinning] mechanism”.

Reviewer #3 (Remarks to the Author):

The authors study the positions of $\sim 1\mu\text{m}$ -wide skyrmions in a CoFeB film that drift by thermal excitation. By examining the residence times, the authors can deduce a 2D map of skyrmion energy versus position, with the implied approximation of the skyrmion by a point particle. Examining the pinned skyrmion at a scale comparable to its diameter, the authors find that the location of the ‘centre-of-mass’ of the skyrmion depends on its diameter, which the authors change by applying a field. From this, the authors claim to observe pinning sites that are dependent on the skyrmion size. The authors also observe the shape of the pinned skyrmion, and observe that the borders of the different-sized skyrmions share a common area. From this the authors deduce that the assumption that pinning is determined by the skyrmion’s centre is false, an assumption that the authors claim to be commonly accepted.

I agree with the authors that problem of skyrmion pinning is interesting, and the observation of skyrmion residence times is an interesting tool to study it.

Author’s reply:

We thank the referee for the careful reading of the paper and the concise summary of our manuscript. We also appreciate the acknowledgement that skyrmion pinning is interesting and that we have employed a useful tool to study this. While referees #1 and #2 recommend publication of our work, the referee has raised a few concerns that we address point-by-point below.

However, I have significant reservations regarding the second and main part of the work. The variation of the skyrmion centre-of-mass with skyrmion size is claimed to be a proof of a variation of pinning sites with skyrmion size. However, the studied skyrmions are larger than the separation between the supposedly different pinning sites. The phenomenon is more straight-forwardly understood as a variation of the shape (and area) of the skyrmion with applied field, pinned at the same site — a conclusion that the authors seem also to share when presenting fig 3. The data of Figs. 2 and 3 do not show, in my opinion, that a “surprisingly strong skyrmion size dependency of the pinning”. Instead, they show a small — relatively to skyrmion diameter — variation of the skyrmion shape. That claim would be supported by a figure in the scale of Fig. 1a for different skyrmion diameters.

Author’s reply:

We thank the referee for the remarks. In the revised version, we now clarify this point: in particular, we highlight in the results section that in Figure 2, the distances between the pinning site 1 and the others are larger than the skyrmion radius. Therefore, the difference between the pinning sites cannot be associated solely with a change in shape. Furthermore, we note that we have carefully verified that changing the size of the skyrmion independently either by only the magnetic field or by temperature leads to skyrmion pinning at the same pinning sites. For this reason, we have clear evidence that the differently pinned skyrmions occur not just due to their shape but also due to their size. Another important remark is that the change in shape is a consequence of the spin configuration attempting a shape configuration that minimizes the energy of the skyrmion, whereas the (equilibrium) size is determined by the applied temperature and/or magnetic field. However, we agree with the referee that

highlighting the magnitude of the size dependence as “extreme” is subjective, and we therefore avoided such descriptions in the revised manuscript.

Another reservation I have is regarding the claim that it is commonly-accepted that the skyrmion's centre is the critical region in the pinning process. No references are given to support that this is a commonly-accepted hypothesis. Indeed, a 'naïve' hypothesis of skyrmion pinning would state exactly the opposite: pinning is a variation of energy with position (dU/dx), therefore the skyrmion centre, where magnetisation is constant ($dM/dx=0$), should not directly influence pinning! It might be common in the literature to describe pinning sites using the skyrmion's centre-of-mass (as in Fig. 1), with the implied approximation of the skyrmion by a particle. However, the data reported in this manuscript does not invalidate this description for the length scales larger than the skyrmion diameter, which is its reasonably expected condition of validity.

Author's reply:

We thank the referee for the remark. The studied scenarios are now listed in great detail including the relevant references. So far in the literature, mostly skyrmions with a continuously rotating magnetization across the skyrmion center have been considered [29-36]. This endows the skyrmion with an increased rigidity. Often, such strong rigidity is extrapolated to larger skyrmions treating them as rigid particles. In this manuscript we show that this extrapolation is not correct. Indeed, for the dipolar-stabilized skyrmion bubbles studied here, the deformation of the skyrmion has a much lower energy cost [44] such that the size and shape of the skyrmion plays a crucial role for their pinning mechanism. We have clarified this point and added additional text and references in the introduction and results section of the revised manuscript.

I find that, although the experiments seem to have been carefully performed and analysed, some of the main claims of the work are not supported by the experimental observations. The study of skyrmion residence times (Fig 1) is interesting. The authors do show that the pinned skyrmion centre-of-mass varies with skyrmion size. However, I do not find that these conclusions are sufficient to warrant publication in Nature Communications.

Author's reply:

We are grateful for the referee's remarks and we have improved the manuscript to emphasize our main results better. In this work we reveal the pinning mechanism for large skyrmions, which we identify by a combination of extensive experimental and numerical efforts. As mentioned by the referee, this is an interesting topic, which we agree with. We also point out that the topic is very timely and has a significant impact, as mentioned by the other referees. The correlation between skyrmion size/shape and the interaction with pinning sites plays an essential role for skyrmion-based devices that depend on both thermally excited and current-induced motion of skyrmions. Moreover, it also provides an explanation on the so-far experimental observation of the motion properties of large skyrmions in multilayers with amorphous magnetic layers [26,27], where large skyrmions are promising candidates for a variety of room temperature skyrmion-based devices.

Therefore, with the relevance of our results as pointed out by the other referees, we believe that the revised manuscript is suitable for publication in *Nature Communications*.

OVERVIEW OVER MAJOR MANUSCRIPT CHANGES

- Explicit explanations of the state-of-the-art including the previous skyrmion center pinning approaches in the rigid particle approximation including references in the introduction
- Clear statement on how the size dependence influences the pinning in the results and discussion section
- New Fig. 3 showing schematically the concept of boundary pinning in a non-flat energy landscape to make our results more accessible
- Additional remarks on why the boundary has to be pinned in our observations of bubble skyrmions both in results and discussion section
- Additional Supplementary Note 2 and Supplementary Figure 2 showing pinning of different size at the same pinning site
- Inclusion of the figure on micromagnetic simulation results (now Figure 5, previously Supplementary Figure 2) into the main text
- Additional Supplementary Note 3 and Supplementary Figure 3 showing exemplarily the arrangement of skyrmions in the micromagnetic simulation
- Corrected specification of the simulation conditions with references to show that our simulation approach employs valid boundary conditions

Reviewers' Comments:

Reviewer #1:

Remarks to the Author:

The authors have made the changes I have suggested and have also made additional changes that other referees have suggested. I think the paper is much improved and can now be published.

Reviewer #2:

Remarks to the Author:

The manuscript has been improved sufficiently to explain their idea of skyrmion boundary pinning. The work might give an opportunity to reconsider the feasibility of skyrmionic devices or inspire how to cope with skyrmion pinning.

This paper should certainly be published in Nature communications.

Reviewer #3:

Remarks to the Author:

I find that the modifications the authors have made clarified some aspects of this work, but have not solved the issues that I found in the first report: that the claim that the pinning sites change significantly with skyrmion size is unsupported by their experimental findings, and that the current state-of-the-art understanding is that the pinning centres interact with the skyrmion centre (it is not). I find, as I did in the first review, that some of the main claims of the work are not supported by the experimental observations. The claim that pinning strength changes with skyrmion size is less novel than what may be understood from the text.

As I wrote in the first report, there are other aspects of the manuscript that are interesting and worth publishing. However, I find that they are insufficient to warrant publication in a wide audience journal such as Nature Communications.

I have detailed below these points. My first report still mostly applies.

[Issue 1]

I maintain what I said in the first report regarding the point-like approximation for pinned skyrmions. The claim that there is a widespread misuse of the approximation of a skyrmion by core position when studying pinning is not justified. Example sentences in the manuscript:

* (L.64) "In particular, the current understanding of pinning is primarily based on theoretical predictions and micromagnetic simulations considering the rigid particle description of skyrmion"

* (L.70) "Depending on the mechanism, defects attracting skyrmion center [...] have been predicted and established the conventional picture of skyrmion center pinning."

* (L.75) "Experimentally, little work is available with in particular strong pinning reported in thin films [28,38-42] where [...] pinning has so far always been considered to be a static property of the sample."

* (L.79) "In particular, the rigid quasi-particle description typically used for skyrmions has neglected their complex spin structure with a core as well as a delineating domain wall boundary both being deformable and does not allow for a description of many of the complex pinning properties observed experimentally"

In fact, it is well-known in the published literature that large pinned skyrmions are complex in shape, and that the point-like approximation is limited in this case. The importance of the extra degrees of freedom of large skyrmions (radius and shape) in pinning has been studied and reported in several articles. It is not true that the treatment of pinning as thus far been reduced to the examination of the skyrmion as a point.

E.g. I. Gross et al., PR Mat. 2 024406 (2018) addresses, both experimentally and numerically, the complex shapes of large pinned skyrmions. Other cited work also show this. In the cited studies where the approximation is used for small skyrmions, the approximation is valid, and its limits are well documented. For example, in ref [23] (Reichardt et al.), when discussing the point-like approximation, the authors warn that:

“Many of the previously studied systems with pinning [...] are composed of relatively stiff particle-like objects in which the internal degrees of freedom are unimportant, making a particle-based treatment of their dynamics appropriate. In contrast, skyrmions can exhibit excitations of internal modes (Beg et al., 2017; Garst et al., 2017; Ikka et al., 2018; Onose et al., 2012) or large distortions (Gross et al., 2018; Litzius et al., 2017; Zeissler et al., 2017) that activate additional degrees of freedom, significantly impacting the statics and dynamics [...]. The uniformity often associated with particle-based models may also not capture the behavior of a skyrmion system well. [...] in some systems, there is considerable dispersion in the size of the skyrmions, making the skyrmion assembly effectively polydisperse (Karube et al., 2018)”

[Issue 2]

The other issue I raised in the first report still remains, concerning the results of Fig 2 and Fig 4 (previous Fig 3). The considered “different pinning sites” can be better explained by a small, relatively to the skyrmion size, change of skyrmion shape, pinned at a same defect.

The authors have responded that the sites are farther separated than the skyrmion radius, but that is not the relevant measure: two pinned configurations at a same site can have centres-of-mass separated by up to twice the radius, even (or especially) if the pinning defect interacts with the skyrmion border. They would still be pinned by the same defect. The sentence in L.221 (“the distance between pinning site 1 and sites 2-4 is of the order of or larger than the skyrmion radius and therefore, the difference between the pinning sites cannot be associated solely with a change in shape but the size is crucial”) is, thus, inaccurate.

Moreover, the authors sometimes identify the pinning site position with the centre of the skyrmion, which is inconsistent with their own careful analysis, that shows that the pinning sites are located at the skyrmion boundary (e.g. L288 “We find that the pinning sites, at which skyrmion centers are located, vary drastically with the skyrmion size.”). This confusion may be a cause of the issue at hand, as it leads to misattributing different pinning sites to changes of the shape of a skyrmion sitting at a same pinning defect.

[Issue 3]

In lines 290-296: the discussion of how pinning can be changed with skyrmion size, in particular how the pinning is most efficient when the skyrmion size matches the pinning distribution length scale, was already experimentally and numerically studied in ref. [40] W. Legrand et al., Nano Lett. 17 (2017). This reference should be cited here.

In general, even ignoring its experimental support, the finding that pinning strength is modified by the skyrmion size is less novel than what may be understood from the text: previously published works already showed (experimentally and numerically) that there is a variation of pinned skyrmion shapes and sizes and, inversely, that pinning is affected by those degrees of freedom.

POINT-BY-POINT REPLY TO REVIEWER COMMENTS

References are numbered identically as in the finalized manuscript.

Reviewer #1 (Remarks to the Author):

The authors have made the changes I have suggested and have also made additional changes that other referees have suggested. I think the paper is much improved and can now be published

Author's reply:

We thank the referee for noting that our paper is “much improved and can now be published”. We are especially grateful for their previous valuable comments helping us to develop the revised manuscript.

Reviewer #2 (Remarks to the Author):

The manuscript has been improved sufficiently to explain their idea of skyrmion boundary pinning. The work might give an opportunity to reconsider the feasibility of skyrmionic devices or inspire how to cope with skyrmion pinning.

This paper should certainly be published in Nature communications.

Author's reply:

We appreciate the referee's comment in addition to the previous helpful suggestions and are grateful for their conclusion that “this paper should certainly be published in *Nature Communications*”.

Reviewer #3 (Remarks to the Author):

I find that the modifications the authors have made clarified some aspects of this work, but have not solved the issues that I found in the first report: that the claim that the pinning sites change significantly with skyrmion size is unsupported by their experimental findings, and that the current state-of-the-art understanding is that the pinning centres interact with the skyrmion centre (it is not). I find, as I did in the first review, that some of the main claims of the work are not supported by the experimental observations. The claim that pinning strength changes with skyrmion size is less novel than what may be understood from the text.

As I wrote in the first report, there are other aspects of the manuscript that are interesting and worth publishing. However, I find that they are insufficient to warrant publication in a wide audience journal such as Nature Communications.

I have detailed below these points. My first report still mostly applies.

Author's reply:

We thank the referee for the detailed review and for considering the topic of our work as interesting. In contrast to the referees #1 and #2, who support publication, the referee is concerned regarding parts of our conclusions. Here we have addressed the issues point-by-point to further clarify some points and rule out some of the referees' proposals. The necessary modifications were added to the finalized manuscript.

[Issue 1]

I maintain what I said in the first report regarding the point-like approximation for pinned skyrmions. The claim that there is a widespread misuse of the approximation of a skyrmion by core position when studying pinning is not justified. Example sentences in the manuscript:

** (L.64) “In particular, the current understanding of pinning is primarily based on theoretical predictions and micromagnetic simulations considering the rigid particle description of skyrmion”*

** (L.70) “Depending on the mechanism, defects attracting skyrmion center [...] have been predicted and established the conventional picture of skyrmion center pinning.*

** (L.75) “Experimentally, little work is available with in particular strong pinning reported in thin films [28,38-42] where [...] pinning has so far always been considered to be a static property of the sample.”*

** (L.79) “In particular, the rigid quasi-particle description typically used for skyrmions has neglected their complex spin structure with a core as well as a delineating domain wall boundary both being deformable and does not allow for a description of many of the complex pinning properties observed experimentally”*

In fact, it is well-known in the published literature that large pinned skyrmions are complex in shape, and that the point-like approximation is limited in this case. The importance of the extra degrees of freedom of large skyrmions (radius and shape) in pinning has been studied and reported in several articles. It is not true that the treatment of pinning as thus far been reduced to the examination of the skyrmion as a point.

E.g. I. Gross et al., PR Mat. 2 024406 (2018) addresses, both experimentally and numerically, the complex shapes of large pinned skyrmions. Other cited work also show this. In the cited studies where the approximation is used for small skyrmions, the approximation is valid, and its limits are well documented. For example, in ref [23] (Reichardt et al.), when discussing the point-like approximation, the authors warn that:

“Many of the previously studied systems with pinning [...] are composed of relatively stiff particle-like objects in which the internal degrees of freedom are unimportant, making a particle-based treatment of their dynamics appropriate. In contrast, skyrmions can exhibit excitations of internal modes (Beg et al., 2017; Garst et al., 2017; Ikka et al., 2018; Onose et al., 2012) or large distortions (Gross et al., 2018; Litzius et al., 2017; Zeissler et al., 2017) that activate additional degrees of freedom, significantly impacting the statics and dynamics [...]. The uniformity often associated with particle-based models may also not capture the behavior of a skyrmion system well. [...] in some systems, there is considerable dispersion in the size of the skyrmions, making the skyrmion assembly effectively polydisperse (Karube et al., 2018)”

Author's reply:

We appreciate the detailed comments about skyrmion pinning mechanisms and observations, which have led to the referee's concern. We agree that large skyrmions are complex in shape and therefore, a complex pinning is expected. However, no thorough analysis has been performed so far and previous work has focused on small nanometer-scale skyrmions. We point out, for example, that in the work of Gross et al. (now added Ref. [30]), a shape and size changes due to pinning is observed but the authors do not discuss the detailed mechanism. The aim of our manuscript is to carefully analyze this mechanism and its consequences on skyrmion size and shape. In response to the comments, we have included the mentioned reference into the finalized version of the manuscript and set the works in context there.

Furthermore, within our work, we do not claim that pinned skyrmions have purely been treated as point-like objects. Note that in the cited investigations [28-38], pinned skyrmions are described as both being centered or located off-center with respect to a single pinning site. Simplifying the complex skyrmion pinning to a point-like pinning site, as considered in previous works, hinders the understanding of the pinning mechanism. We show that even though an accumulation of skyrmion center occurrences can still be associated with a pinning site as defined in the manuscript, the energy landscape in the vicinity of the pinning site plays a crucial role.

Concerning ref. [23], the large deformations and excitations discussed were not observed in our experimental data. We argue that excitation modes do not play a role on the time scale investigated in the manuscript and our observations reveal only static smooth skyrmion deformations within a rigid energy landscape.

[Issue 2]

The other issue I raised in the first report still remains, concerning the results of Fig 2 and Fig 4 (previous Fig 3). The considered “different pinning sites” can be better explained by a small, relatively to the skyrmion size, change of skyrmion shape, pinned at a same defect.

The authors have responded that the sites are farther separated than the skyrmion radius, but that is not the relevant measure: two pinned configurations at a same site can have centres-of-mass separated by up to twice the radius, even (or especially) if the pinning defect interacts with the skyrmion border. They would still be pinned by the same defect. The sentence in L.221 (“the distance between pinning site 1 and sites 2-4 is of the order of or larger than the skyrmion radius and therefore, the difference between the pinning sites cannot be associated solely with a change in shape but the size is crucial”) is, thus, inaccurate.

Moreover, the authors sometimes identify the pinning site position with the centre of the skyrmion, which is inconsistent with their own careful analysis, that shows that the pinning sites are located at the skyrmion boundary (e.g. L288 “We find that the pinning sites, at which skyrmion centers are located, vary drastically with the skyrmion size.”). This confusion may be a cause of the issue at hand, as it leads to misattributing different pinning sites to changes of the shape of a skyrmion sitting at a same pinning defect.

Author’s reply:

We thank the referee for this comment. To clarify this point further, we note that for the experimental data shown in the manuscript, the skyrmions pinned by the boundary do not show significant translational motion but are rather recurring statically at specific positions. Therefore, we can assign a position for the center of mass where the skyrmion is pinned. This however does not rule out that for different configurations of the energy landscape, the pinning on the border may lead to different possible positions for the center of the skyrmion (e.g., just a small region of the border of the skyrmion is pinned). We especially leave room for several sections of the skyrmion boundary being pinned simultaneously for one specific skyrmion, which is now explained explicitly in the revised discussion. We also emphasize that, the pinning regions are localized and separated from each other, which allows us to argue that the skyrmions are not pinned at a single point as suggested by the referee.

[Issue 3]

In lines 290-296: the discussion of how pinning can be changed with skyrmion size, in particular how the pinning is most efficient when the skyrmion size matches the pinning distribution length scale, was already experimentally and numerically studied in ref. [40] W. Legrand et al., Nano Lett. 17 (2017). This reference should be cited here.

In general, even ignoring its experimental support, the finding that pinning strength is modified by the skyrmion size is less novel than what may be understood from the text: previously published works already showed (experimentally and numerically) that there is a variation of pinned skyrmion shapes and sizes and, inversely, that pinning is affected by those degrees of freedom.

Author's reply:

We thank the referee for the remark. We notice that previous literature focuses on an energy landscape dependence on the pinning of skyrmions for skyrmions about the same size as the grains of the sample [22,29]. Here, however, we perform a thorough experimental analysis of this behavior, going beyond grains with sizes compared to the skyrmion size, and showing that this process also leads to a previously elusive size dependent pinning mechanism. We have added this comment to the introduction of the finalized manuscript to underline the novelty of our findings.

Clarifying or ruling out concerns raised by the referee and knowing that the other two referees support publication, we believe that the finalized manuscript is ready for publication in *Nature Communications*.

Reviewer #3:

Remarks to the Author:

I have to maintain mostly the assessment of my previous report.

The modifications the authors have made have not solved the issues of my first two reports, especially regarding the claim that the pinning sites change significantly with skyrmion size, which is unsupported by their experimental findings.

Also, this claim is less novel than what may be understood from the text, although some added references now address better previous works. I find that the description of the current state-of-the-art was slightly improved, although it still largely transmits the (incorrect) notion that the current understanding is that the pinning occurs at the skyrmion's centre.

In particular, regarding the finding that different-sized skyrmions pin at different sites, I have carefully read the authors response before re-reading the manuscript. I find again that the data of Figs. 2 and 4 do not substantiate the claim that the skyrmions are pinned by distinct pinning sites. As I wrote previously, the considered "different pinning sites" can be better explained by a small (relatively to the skyrmion size) change of skyrmion shape, pinned at a same defect.

I will try to clarify the issue.

I do agree that the data shows that the centres of pinned skyrmions are clustered together in distinct clusters depending on their size. These clusters are separated by less than the skyrmion diameter (1 to 2 μm separation compared to ~ 2.5 and $\sim 3.5\mu\text{m}$ for the diameter). Therefore, both large and small skyrmions partially cover a same "shared area" of the film. Therefore, the data is perfectly compatible with skyrmions being pinned at a same defect (or defects) in this "shared area", with skyrmions of different sizes adopting different shapes, and so with their centres sitting at different positions.

In their response, the authors write "We also emphasize that, the pinning regions are localized and separated from each other, which allows us to argue that the skyrmions are not pinned at a single point as suggested by the referee." The clusters of skyrmion centres are separated between small and larger skyrmions, but, again, the separation being smaller than the diameter, this does not exclude a common pinning site in the "shared area".

This is not just language. For example, the claim "...by varying the field and thus the size, we can switch on and off certain pinning sites and thus tune the pinning on-the-fly" is not proven by the data of Fig. 2. It is perfectly possible that that what is being "switch off" are certain configurations: the larger skyrmions are still being pinned by the defect, only their centres are sitting at a different, but neighbouring, point. If that is the case, applying a field to change the skyrmion size will result in a different shape of the pinned skyrmion but it will not avoid its pinning.

As such, I still disagree with the sentence in the manuscript "However, the distance between pinning site 1 and sites 2-4 is of the order of or larger than the skyrmion radius and therefore, the difference between the pinning sites cannot be associated solely with a change in shape but the size is crucial" .

As I wrote in the first and second reports, there are other aspects of the manuscript that are interesting and worth publishing. However, I find that they are insufficient to warrant publication in a wide audience journal such as Nature Communications.

POINT-BY-POINT REPLY TO REVIEWER COMMENTS

*References are numbered identically as in the revised manuscript. Ref. 48 was replaced by the now published version (Horenko, I., Rodrigues, D., O’Kane, T. & Everschor-Sitte, K. Scalable computational measures for entropic detection of latent relations and their applications to magnetic imaging. *Commun. Appl. Math. Comput. Sci.* **16**, 267–297 (2021).)*

Reviewer #1 (Remarks to the Author)

Referee 3 has made some points that I think the authors have addressed. The referee 3 has pointed out that there are already some works considering non-point like approaches to skyrmion pinning effects. The authors have added some references on that. I would add that pinning in skyrmions is in general an open question and it is probably true that there are different pinning regimes where in some cases the particle like approach with the skyrmion being pinned in the center of site works, while in other cases it is the edge or interface of the skyrmions. This will depend on the size, and stiffness of the skyrmion and details of the pinning. In any case pinning of skyrmions will be a relevant issue for many applications. In light of the importance of the problem, and and this being a experimental investigation I think the paper can be published.

Author’s reply:

We thank the referee for noting that our paper is worth being published in *Nature Communications*. We are especially grateful for all their valuable comments helping us to develop the revised manuscript. As suggested, we have now explicitly added to the manuscript that skyrmion pinning is “in general an open question” to the introduction and furthermore emphasized in the discussion that the particle-like pinning description may work, but “we find that in other cases it is the skyrmion boundary which is pinned – depending on the skyrmion size and stiffness as well as the energy landscape”.

Reviewer #3 (Remarks to the Author)

I have to maintain mostly the assessment of my previous report.

The modifications the authors have made have not solved the issues of my first two reports, especially regarding the claim that the pinning sites change significantly with skyrmion size, which is unsupported by their experimental findings.

Also, this claim is less novel than what may be understood from the text, although some added references now address better previous works. I find that the description of the current state-of-the-art was slightly improved, although it still largely transmits the (incorrect) notion that the current understanding is that the pinning occurs at the skyrmion's centre.

Author’s reply:

We thank the referee for the careful study of our manuscript and their review. We appreciate that our manuscript is considered improved and thank for the previous comments helping us with this development. In contrast to the reviewer #1, who thinks that “the authors have addressed” the points raised and supports publication in *Nature Communications*, the referee argues that we “transmit the (incorrect) notion that the current understanding is that the pinning occurs at the skyrmion's centre”. For the concerns regarding the presentation of our results, we are willing to address the comments point by point below.

In particular, regarding the finding that different-sized skyrmions pin at different sites, I have carefully read the authors response before re-reading the manuscript. I find again that the data of Figs. 2 and 4 do not substantiate the claim that the skyrmions are pinned by distinct pinning sites. As I wrote previously, the considered “different pinning sites” can be better explained by a small (relatively to the skyrmion size) change of skyrmion shape, pinned at a same defect.

I will try to clarify the issue.

I do agree that the data shows that the centres of pinned skyrmions are clustered together in distinct clusters depending on their size. These clusters are separated by less than the skyrmion diameter (1 to 2 μm separation compared to ~ 2.5 and $\sim 3.5\mu\text{m}$ for the diameter). Therefore, both large and small skyrmions partially cover a same “shared area” of the film. Therefore, the data is perfectly compatible with skyrmions being pinned at a same defect (or defects) in this “shared area”, with skyrmions of different sizes adopting different shapes, and so with their centres sitting at different positions.

In their response, the authors write “We also emphasize that, the pinning regions are localized and separated from each other, which allows us to argue that the skyrmions are not pinned at a single point as suggested by the referee.” The clusters of skyrmion centres are separated between small and larger skyrmions, but, again, the separation being smaller than the diameter, this does not exclude a common pinning site in the “shared area”.

Author’s reply:

We thank the referee for clarifying their understanding of the pinning mechanism. Indeed, their explanation describes can describe some of our observations, however the conclusion that the exact same defects can be responsible for the observed behavior is clearly not correct. As already examined in the manuscript (in *Results/Single Skyrmion Pinning*: “We notice however that for such large skyrmions of micrometer size, the core is homogeneously magnetized. Thus, the translation of the homogeneous core corresponds to a zero mode and cannot influence the pinning site”), we can exclude defects which are pinning parts of the skyrmion core from being responsible for the observed pinning. Since the core is homogeneous and of micrometer size (i.e. orders of magnitudes larger than the domain wall at the skyrmion boundary), skyrmion positions would not be clustered on such short length scales in that case, but rather evenly distributed in this “shared area”.

However, we agree that even for different positions at which skyrmion centers accumulate (which is called pinning sites in the manuscript consistently, c.f. in the discussion: “We find that the pinning sites, at which skyrmion centers are located, vary drastically with the skyrmion size”), the skyrmions may share some of the defects. In Fig. 4a, we observe that the skyrmion boundary follows complex contours, which are nevertheless characteristic and recurring for certain skyrmion sizes. Parts of the contours are still coinciding which is conclusive evidence that the boundary is pinned. To clarify potential misunderstandings, we want to stress that all shown contours are observed at the same magnetic field value. We have added the field value to the figure caption to avoid such misunderstanding and demonstrate that the area/size is not manipulated by a magnetic field. However, to change between the contours, different parts of the contour have to be moved (e.g. the left part is different for the red and green contour while the right part coincides; vice versa, for the red and black ones the right part is different). That is, pinning on one side while allowing for an area extension in the other direction (as observed for different fields in Supplementary Fig. 2) is not sufficient to explain this behavior. This entails the notion that different or additional defects have to be taken into account.

We agree with the referee that “the data shows that the centres of pinned skyrmions are clustered together in distinct clusters depending on their size”, which is one of the main messages of our paper according to what we have just explained in the above. Finally, we emphasize that for this, the whole complex energy landscape is therefore decisive for how a particular skyrmion is pinned.

This is not just language. For example, the claim “...by varying the field and thus the size, we can switch on and off certain pinning sites and thus tune the pinning on-the-fly” is not proven by the data of Fig. 2. It is perfectly possible that that what is being “switch off” are certain configurations: the larger skyrmions are still being pinned by the defect, only their centres are sitting at a different, but neighbouring, point. If that is the case, applying a field to change the skyrmion size will result in a different shape of the pinned skyrmion but it will not avoid its pinning.

As such, I still disagree with the sentence in the manuscript “However, the distance between pinning site 1 and sites 2-4 is of the order of or larger than the skyrmion radius and therefore, the difference between the pinning sites cannot be associated solely with a change in shape but the size is crucial”.

Author’s reply:

We thank the referee for this comment, which we would like to answer in the context of the above reply. We agree with the referee that this could be a possibility as shown in Supplementary Fig. 2. However, the data shown in Fig. 2 in the main manuscript reveals “distinct clusters depending on their size” providing strong evidence that indeed there is a size dependence as described above. We note that some positions are however not accessible anymore for skyrmions of certain sizes (such as position 1 in Fig. 4a, which is hosts only the smallest of the observed skyrmions). With this, the effective localization of the center is still mediated by the skyrmion size. We nevertheless agree to change the mentioned sentence in the manuscript to express our findings in a better understandable way.

As I wrote in the first and second reports, there are other aspects of the manuscript that are interesting and worth publishing. However, I find that they are insufficient to warrant publication in a wide audience journal such as Nature Communications.

Author’s reply:

We thank the referee for noting that there are “aspects of the manuscript that are interesting and worth publishing” and are grateful for their comments helping us to improve our manuscript over the course of the three review iterations. After clarifying and explaining the raised concerns and considering the positive reports from reviewers #1 and #2, we feel confident that our manuscript is suitable for publication in *Nature Communications*.